# Can an Observational Gait Scale Produce a Result Consistent with Symmetry Indexes Obtained from 3-Dimensional Gait Analysis?: A Concurrent Validity Study

**DOI:** 10.3390/jcm9040926

**Published:** 2020-03-28

**Authors:** Agnieszka Guzik, Mariusz Drużbicki, Lidia Perenc, Justyna Podgórska-Bednarz

**Affiliations:** Institute of Health Sciences, Medical College, University of Rzeszów, 35-959 Rzeszów, Poland; mdruzb@ur.edu.pl (M.D.); lidiaiadam.perenc@wp.pl (L.P.); j.e.podgorska@googlemail.com (J.P.-B.)

**Keywords:** Stroke, gait analysis, symmetry index

## Abstract

To investigate whether a simple observational tool may be a substitute to the time-consuming and costly 3-dimensional (3D) analysis, the study applied the Wisconsin Gait Scale (WGS), enabling assessment which is highly consistent with 3D gait parameters in patients after a stroke. The aim of this study was to determine whether, and to what extent, observational information obtained from WGS-based assessment can be applied to predict results of 3D gait analysis for selected symmetry indicators related to spatiotemporal and kinematic gait parameters. Fifty patients at a chronic stage of recovery post-stroke were enrolled in the study. The spatiotemporal and kinematic gait parameters were measured using a movement analysis system. The symmetry index (SI), was calculated for selected gait parameters. The patients’ gait was evaluated by means of the WGS. It was shown that stance % SI, as well as hip and knee flexion-extension range of motion SI can most effectively be substituted by WGS-based estimations (coefficient of determination exceeding 80%). It was shown that information acquired based on the WGS can be used to obtain results comparable to those achieved in 3D assessment for selected SIs of spatiotemporal and kinematic gait parameters. The study confirms that observation of gait using the WGS, which is an ordinal scale, is consistent with the selected aims of 3D assessment. Therefore, the scale can be used as a complementary tool in gait assessment.

## 1. Introduction

Despite the fact that 3-dimensional gait analysis (3DGA) is the most accurate method enabling gait assessment and is recognized as a key outcome measure by gait researchers and healthcare professionals [1], for many facilities it is inaccessible due to the considerable costs involved. Consequently, observational gait analysis is more commonly used in practice, as it is a fast, simple and inexpensive method [2]. 

The Wisconsin Gait Scale (WGS), which is an observational tool enabling the assessment of gait quality in patients after a stroke, was developed by Rodriguez et al. [3]. The WGS consists of four subscales and assesses 14 observable gait parameters occurring during the consecutive gait phases, i.e., stance phase, toe off, swing phase, and heel strike of the affected leg. The WGS has been shown to be accurate and reliable in patients post-stroke, therefore it can effectively be used to evaluate progress in gait rehabilitation after a stroke [3,4,5,6,7,8]. Originally, the WGS was intended to be used as a tool to enable the assessment of effects achieved during a home-based gait training program designed for patients at a chronic stage of recovery post-stroke [3]. The authors subsequently tested the validity of the WGS and showed its effectiveness as an instrument of gait measurement, making it possible to compare outcomes [3]. Similarly, Yaliman et al. [7] and Wellmon et al. [8] established that the WGS is highly reliable, and they suggested it may be used as an objective tool for recording results of an observational gait analysis after a stroke. Estrada-Barranco et al. [4] showed that the correlation of the WGS with the balance and functionality scales in stroke patients at subacute and chronic stages was excellent. Pizzi et al. [6] reported that the WGS may be used effectively to evaluate qualitative changes in gait patterns presented by patients with post-stroke hemiparesis, and to assess changes observed during the course of long-term rehabilitation. The tool may be effectively applied effectively when it is necessary to perform targeted and standardized measurements of hemiplegic gait in order to accurately modify rehabilitation in line with the results of monitoring [6].

The respective parts of the WGS focus on spatiotemporal (subscale one) and kinematic parameters of gait (subscale one, two, three, and four). Video recordings are analyzed by an examiner and based on that, a score is assigned for the performance of the gait pattern [3]. The specific items of the scale focus on positions assumed by parts of the lower extremities and joints during the gait cycle, while taking into account both the affected and the unaffected leg, which are then compared. The WGS makes it possible to identify basic defects in the gait pattern, such as asymmetries observed in step length, duration of gait phases, and body weight shifting, as well as impaired mobility in the joints of the paretic leg. As a result, the WGS provides a qualitative description of gait, mainly in terms of gait asymmetry [3,6,8]. According to Pizzi et al. [6] and Lu et al. [9] the observational WGS measures changes occurring in hemiplegic gait, which is reflected in the inter-limb symmetry during movement, weight shift and weight-bearing during the stance and swing phases, as well as the ankle, knee, and hip kinematics, balance/guardedness, and the use of assistive devices. Wellmon et al. [8] argued that the WGS provides important complementary information on patients’ walking performance, making it possible to identify reductions in spatial and temporal gait asymmetries, which are indicative of recovery and better walking stability in patients with stroke.

The motivation for our research was provided by the studies published by Patterson et al. [10,11] who emphasized that nowadays increasing attention is paid to symmetry as a basic gait characteristic, particularly in patients after a stroke. These researchers also pointed out that, from clinical viewpoint, gait symmetry is of essential importance as it may be linked to various negative consequences, including inefficiency, poor balance control, loss of bone mass density in the paretic lower limb, as well as increased risk of musculoskeletal injury to the unaffected lower limb [11]. Therefore, it is important to identify and understand any existing asymmetries, found in various gait parameters. In line with this requirement, higher WGS scores are indicative of greater gait asymmetry [3,6,8]. A number of inter-limb deviations account for symmetry or asymmetry, which is a significant feature of gait. Importantly, in the related literature the terms symmetry and asymmetry are commonly used when referring to the same parameters [12]. Healthy individuals generally present symmetrical gait, with only minor deviations. Hence, it seems that normal and pathological gait patterns are effectively differentiated by the characteristic of asymmetry, or lack of symmetry [10]. Another reason for selecting this specific subject matter for our research lies in the fact that many studies have already reported findings on the feasibility of the WGS as a tool for observational assessment of gait asymmetry in patients post-stroke [3,6,7,8].

As it reflects similarities in spatiotemporal and kinematic parameters of the right and left lower limb, symmetry is a significant measure of gait assessment, and can more effectively describe post-stroke gait mechanisms compared to conventional measures such as velocity. Furthermore, it may provide important information to be taken into account by clinicians during treatment decisions [13]. In the case of patients with hemiplegia, asymmetries are observed in stride time and length, as well as the timing in the single-limb support phase on the paretic and non-paretic limbs [14,15,16]. Furthermore, temporal and spatial asymmetries are commonly found in a number of gait measures in comparative assessments of the hemiparetic and the unaffected side [11]. There may be inequalities in step length resulting from efforts aimed at reducing weight bearing on the paretic limb. It is also possible to observe shorter stance time on the paretic limb [17], increased base of support during double stance, and difficulty clearing the paretic foot during the swing phase [15]. Other researchers have reported differences in trunk alignment and limited hip and knee flexion and extension [18,19]. Quality of gait, reflected by symmetry in step length and duration of gait phases, tends to change towards greater asymmetry at later stages post stroke [13]. Given this, the aim of a gait rehabilitation program in patients with a stroke is generally to improve gait symmetry [20]. The WGS enables observational assessment of gait symmetry for factors such as stance time (temporal symmetry), step length (spatial symmetry), and hip and knee range of motion (kinematic symmetry) [3,8].

The aim of this study was to determine whether, and to what extent, observational information about gait symmetry obtained from a WGS-based assessment can be used as a complementary tool that can be applied to predict results of 3D gait analysis for selected symmetry indicators related to spatiotemporal and kinematic gait parameters. 

## 2. Materials and Methods 

### 2.1. Study Protocol

This concurrent validity study was conducted among patients treated at the Rehabilitation Clinic of Provincial Hospital No. 2 in Rzeszow, Poland. All qualifying patients were fully informed about the procedure and signed informed consent to participate in the study. The research protocol was approved by the local Bioethics Commission of the Medical Faculty (5/2/2017), and the study was registered at the Australian New Zealand Clinical Trials Registry (ACTRN12617000436370). 

### 2.2. Participants

The study included 50 patients at a chronic period after a stroke (18 females, 32 males) with a mean age 60.9 ± 11.2 years (range 30–75) and mean time from a stroke 42 months (range 8–120). Of the 50 patients, 15 had right hemisphere lesions, and 35 had left hemisphere lesions. The inclusion criteria were: Ischemic stroke; time from a stroke at least 6 months; independent walking (use of a cane, crutches or Ankle Foot Orthosis (AFO) orthosis was permitted); and Brunnström recovery stage 3–4. Strokes were confirmed by computed tomography or magnetic resonance imaging. The exclusion criteria were: Unstable hemodynamic state; peripheral vascular disease; cognitive impairment (Mini-Mental State Examination <24); lower limb contractures; difference in the length of the lower limbs exceeding 2 cm; osteoarthritis impairing gait; and other orthopedic, rheumatic and neurologic comorbidities impairing ambulation. Table 1 describes the laboratory characteristics of the patients’ gait. 

### 2.3. Outcome Measures

#### 2.3.1. Primary Outcome Measure: Spatiotemporal and Kinematic Parameters of Gait

The patients’ walking abilities were assessed at the Laboratory of Biomechanics of the Institute of Physiotherapy, University of Rzeszów. A 3-dimensional gait analysis was carried out using the SMART system (6 cameras, 120 Hz), manufactured by BTS Bioengineering (BTS Bioengineering, Milan, Italy). The internal protocol of the system (Helen Hayes (Davis) Marker Placement) was applied in selecting locations for reference markers; these were placed on the sacrum, pelvis (anterior posterior iliac spine), femur (lateral epicondyle, great trochanter, and in lower one-third of the shank), fibula (lateral malleolus, lateral condyle end in lower one-third of the shank), and foot (metatarsal head and heel) [21]. The patients were asked to walk at a comfortable self-selected speed, at a distance of 10 m, and were allowed to use auxiliary equipment, such as canes and elbow crutches, during the examination. During one trial, six passes of the patient were recorded. At the next stage spatiotemporal and kinematic parameters were calculated with the use of tracker and analyzer programs (BTS Bioengineering) by averaging the results to a single session. The analysis took into account: Spatiotemporal parameters including stance time (s), stance phase (% of gait cycle), step length (m), stride length (m) of the paretic and of the non-paretic limb; and kinematic parameters including hip flexion/extension range of motion (Hip FE ROM) and knee flexion/extension range of motion (Knee FE ROM) of the paretic and of the non-paretic limb. 

The video recording and 3D recording were carried out concurrently. Positioning of the two video cameras (BTS Vixta, BTS Bioengineering Corp., Milan, Italy), working in synchronicity, was selected in such a way as to obtain images in the frontal and the sagittal plane. The walking path was 10 m long. One camera was set in line with the direction of the gait in the frontal plane, the other camera, recording sagittal plane view, was positioned halfway along the walking path, 2 m away from the path. The cameras were programmed to allow visualization of three walking trials examining the paretic and the non-paretic sides for a total of six ambulation trials. The subjects were asked to walk the specified distance at a self-selected (comfortable) speed, with the support of orthopedic aids if used on a regular basis.

#### 2.3.2. Secondary Outcomes

The recordings and WGS-based gait assessment were reviewed and interpreted by a physical therapist with over 10 years of experience in working with patients post-stroke, and with expertise using the WGS and interpreting the scores. A physical therapist reviewed the gait assessments via video. Based on the analysis of the video recordings, the evaluator assigns scores for the performance of the gait pattern. The WGS system allows one to assess 14 observable gait parameters. The five initial items relate to affected leg stance phase: (1) use of hand-held assistive device; (2) stance time on the affected side; (3) step length on the unaffected side; (4) weight shift to the paretic limb with/without assistive device; and (5) stance width. The next part of the scale relates to the affected leg toe-off phase and comprises two items: (6) guardedness (pause before advancing the paretic leg); and (7) affected leg hip extension (observation of gluteal crease). The third subscale covers the affected leg swing phase, and comprises six items: (8) external rotation during initial swing; (9) circumduction at mid-swing (path of the paretic foot); (10) hip hiking at mid-swing; (11) knee flexion from toe off to mid-swing; (12) toe clearance; and (13) pelvic rotation at terminal swing. The final subscale relates to the affected leg heel strike and comprises only one item: (14) initial foot contact of the affected leg. The total score, in the range of 13.35–42 points, is calculated for all the items. The points assigned to items 2–10 and 12–14 are added up. Responses to items 1 and 11 are weighted by 3/5 and 3/4, respectively and then the points are added to the total score. Higher scores correspond to poorer overall walking performance and more visible gait deviations [3.8]. Good intra- and inter-rater reliability of the WGS in patients post-stroke has been demonstrated by a number of studies [6,7,8,9,22,23].

### 2.4. Data Analyses

The analyses took into account six gait symmetry indexes calculated based on 3D assessment involving 50 patients. In the current study, the authors decided to apply symmetry indexes (SI), based on 3D gait parameters, to compare the results of equipment-based 3D assessment to those obtained using observational gait analysis performed using the subjective WGS. The measurement of gait symmetry applied the most commonly used method, the absolute index proposed by Robinson [24], which has been used by numerous researchers assessing patients after a stroke [11,25,26,27] and other cohorts. This index has been used to analyze symmetry in long distance runners, and in healthy subjects, as well as those with leg length discrepancies and amputations [28]. It is calculated as a quotient of the absolute difference between the measures for both legs and the mean of these measures, multiplied by 100. The absolute value of the difference between the affected and the unaffected side was taken into account as gait defects are reflected by the disparity between the results identified for both legs, regardless of whether a higher result is found for the right or the left leg. Given the method applied in determining the symmetry index (it can only assume positive values), we can expect that its distribution will be concentrated around values approaching zero. For the ideal symmetrical gait condition, the symmetry index should be zero, reflecting perfectly symmetrical gait pattern. Higher symmetry index values correspond to greater gait asymmetry [24,27]. SI was calculated from the formula:$$SI=\frac{2\;\left(\mathrm{xn}\;-\;\mathrm{xi}\right)}{\left(\mathrm{xn}\;+\;\mathrm{xi}\right)}\;\cdot \;100\;\;.\;$$
where: *xn* is the value of the variable obtained from the non-paretic limb; and *xi* is the value of the corresponding variable obtained from the paretic limb [16].

The results of measurements obtained in 3D assessment for the non-paretic and paretic limb as well as the specific SIs were presented in the form of descriptive statistics and histograms. Analysis of the correlations between 3D symmetry indexes and the specific components, as well as the total score, of the WGS-based assessment is shown in the form of a correlation matrix presenting values of Spearman’s rank correlation coefficients. The strength of all the correlations were interpreted as: 0.3 ≤ ||R| < 0.5 low correlation; 0.5 ≤ |R| < 0.7 moderate correlation; 0.7 ≤ ||R| < 0.9 strong correlation; 0.9 ≤ ||R| < 1 very strong correlation [29].

During the next stage, regression analysis was applied to investigate whether a simple observational tool may be a substitute to the time-consuming and costly 3D analysis. Symmetry indexes based on 3D assessment of the relevant parameters were adopted as dependent variables for the specific models, while scores in items 1–14 of the WGS were applied as independent variables. Subsequently, stepwise regression with forward selection was applied to find a model combining two desired characteristics: It would only contain statistically significant factors and would most successfully describe variability of the indexes. Determined based on such calculations, a regression model for each 3D symmetry index allows to estimate its value using selected WGS scores [30]. 

Statistical significance was assumed to be *p* < 0.05. Statistical analyses were conducted with the use of Statistica 10.0 program (StatSoft, Cracow, Poland).

## 3. Results

### 3.1. Gait Symmetry Indexes

Symmetry indexes were determined for the six gait parameters taken into account in the study (Table 2, Figure 1). Notably, there are clearly visible differences in median values for most symmetry indexes as well as very high maxima. This reflects the fact that for majority of the subjects, SIs assume low or average values, while in a few outstanding cases they are high or extremely high. It can be noticed that the SI is characterized by significant right-side asymmetry, which is shown by the skewness coefficient approaching or even exceeding 1 (Table 2).

The analysis of the SI values identified for the specific parameters and the related graphic presentation suggest that a considerably greater relative differences between the two limbs are found in the parameters of Step Length (s), Hip FE ROM, and Knee FE ROM (Figure 1).

### 3.2. Correlations Between Symmetry Indexes and Scores in the WGS

It was shown that 3D symmetry indexes, related to Stance Time (s), Stance %, Hip FE ROM, and Knee FE ROM may be described with fairly high accuracy using item questions of the WGS (the identified correlations were strong 0.7 ≤ |R| < 0.9 or very strong 0.9 ≤ |R| < 1). The weakest correlation to WGS scores was found in the case of Step Length (m) SI (0.3 ≤ |R| < 0.5) (Table 3). The subsequent stage involved construction of regression models.

### 3.3. Regression Models Describing 3D Symmetry Indexes Using WGS Scores

Table 4 presents the determined regression models which can be used to estimate corresponding symmetry indexes based on selected components of the WGS. Feasibility of substituting symmetry indexes with estimations based on the WGS is rather high for such rates as Stance % SI, Hip FE ROM SI, and Knee FE ROM SI. However, in the case of the remaining three symmetry indexes, the values determined using appropriate equations did not reflect the distribution of 3D measurements as effectively. The poorest results were found for Step Length (m) SI. The coefficient of determination at a level of approximately 24% shows that modelling of the index using WGS scores does not produce satisfying results.

## 4. Discussion

The present study has a practical dimension as it confirms that the WGS is a valuable gait assessment tool and in the circumstances when the use of costly objective methods is not feasible, the WGS may effectively be applied as a diagnostic instrument to perform evaluation of post-stroke gait pattern asymmetry. 

Hsu et al. [31] demonstrated that individuals after a stroke frequently present asymmetric gait patterns, in terms of both temporal and spatial parameters. Patterson et al. [13] indicated that quality of gait, as measured by spatial and temporal symmetry, appears to deteriorate at a later period following a stroke. Bensoussan et al. [18], Balaban and Tok [32] reported that asymmetry in hemiplegic gait also applies to kinematic parameters. Gait pattern asymmetry may contribute to postural instability, musculoskeletal disorders, and ineffective gait leading to greater energy expenditure [33,34,35]. The present findings provide evidence for gait pattern asymmetry observed in patients who have experienced a stroke and are in a chronic stage of recovery. Analysis of SI values related to the specific parameter showed that considerably greater relative differences between the two limbs were found in Step Length (s), Hip FE ROM, and Knee FE ROM. This suggests that spatiotemporal and kinematic gait pattern asymmetry in patients after a stroke is a serious problem; the absolute values of gait parameters, particularly step length, are valuable, above and beyond the symmetry indices. Without an age-matched healthy control group tested in the same laboratory conditions (with the same 3D gait equipment), it is difficult to be certain what disabilities this research fails to detect. Therefore, researchers continue to look for tools enabling effective observational gait pattern assessment after a stroke [22].

In order to answer the question formulated in the purpose of the study, and determine whether and to what extent observational information about gait symmetry obtained from WGS-based assessment can be used as a complementary tool and can be applied to predict results of 3D gait analysis for selected symmetry indicators related to spatiotemporal and kinematic gait parameters, the current analyses were designed to examine correlations of 3D symmetry indexes and the specific items of (as well as the total score in) the WGS. It was shown that 3D symmetry indexes, related to Stance Time (s), Stance %, Hip FE ROM, and Knee FE ROM may be described with fairly high accuracy using item questions of the WGS (0.7 ≤ |R| < 0.9; 0.9 ≤ |R| < 1). This initial finding provided a rationale for the assumption that a combination of selected WGS items may enable even more accurate estimation of symmetry indexes for 3D parameters. Therefore, the next stage involved building regression models, where gait symmetry indexes for selected 3D parameters were used as dependent variables and specific components of WGS (items 1–14) were applied as independent variables. Given the fact that values of dependent variables do not have to be significantly related to all independent variables applied in a regression model, the analyses were designed to find an optimum model that would only contain statistically significant factors and would most effectively describe variability of the dependent characteristic. A formula determined as a result of this procedure allows to estimate a 3D symmetry index based on WGS scores. It was shown that Stance % SI, Hip FE ROM SI, and Knee FE ROM SI can most effectively be substituted by WGS based estimations (coefficient of determination exceeding 80%). In the case of Stance Time (s) SI and Stride Length (m) SI, the WGS values determined using appropriate equations did not reflect the distribution of 3D measurements as effectively. The least satisfying results were found in the case of Step Length (m) SI (coefficient of determination at a level of approximately 24%). We believe that the poor result related to Step Length may be linked with the fact that video recording and processing were carried out only at a basic level.

Besides the traditional 3-dimensional motion capture, a number of advanced objective tools for gait analysis are available at present, e.g., GaitRite or Inertial Measurement Units. However, they are all costly and require technical expertise and specialist equipment. The current findings are not intended to undermine the importance of the objective gait assessment methods, which provide gold standard in this area; they only show that the WGS enables accurate assessment of gait, corresponding to the objective 3D gait measures. The findings suggest that in situations when costly equipment-based gait assessment systems are not accessible to medical facilities for various reasons, the regression models described here may be helpful for physicians and physiotherapists, enabling rough estimation of selected 3D symmetry indexes based on WGS scores. The study shows that this simple, easy, and time-effective scale can be used as a complementary tool in gait assessment and can be recommended as an alternative tool when 3DGA is unavailable. A limitation of the present study lies in the fact that traditional 3-dimensional motion capture was the only objective gait analysis method applied. Therefore, it is necessary to continue the research taking into account other advanced tools for clinical gait analysis.

## 5. Conclusions

It was shown that information acquired based on the WGS can be used to obtain results comparable to those achieved in 3D assessment for selected SIs of spatiotemporal and kinematic gait parameters. 3D symmetry indexes related to the parameters of Stance %, Hip FE ROM, and Knee FE ROM can be fairly accurately described using item questions of WGS. In the case of Step Length (m) SI, the values of WGS determined using appropriate equations reflected the distribution of 3D measurements least effectively. The study confirms that observation of gait using the WGS, which is an ordinal scale, is consistent with the selected aims of 3D assessment, therefore, the scale can be used as a complementary tool in gait assessment.

## Figures and Tables

**Figure 1 jcm-09-00926-f001:**
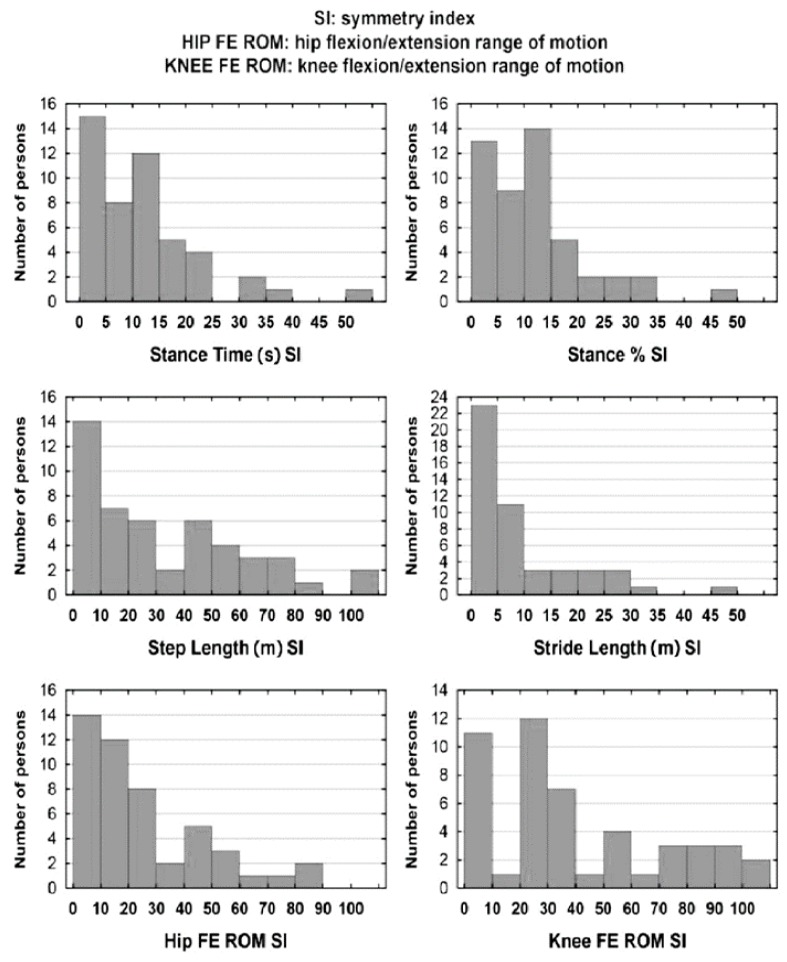
SI distribution for the specific gait parameters. SI, symmetry index; FE, flexion/extension; ROM, range of motion.

**Table 1 jcm-09-00926-t001:** Characteristics of spatiotemporal and kinematic gait parameters in the study group.

3-Dimensional Gait Parameters	$\bar{x}$	*s*	95% C.I.	Med	Min	Max
Paretic limb						
Stance Time (s)	1.11	0.33	(1.01, 1.21)	1.02	0.62	1.86
Stance %	0.66	0.06	(0.64, 0.68)	0.67	0.54	0.79
Step Length (m)	0.24	0.11	(0.20, 0.27)	0.21	0.07	0.56
Stride Length (m)	0.60	0.24	(0.53, 0.67)	0.55	0.22	1.17
Hip FE ROM	27.4	8.2	(25.0, 29.8)	26.3	10.9	47.7
Knee FE ROM	28.8	11.8	(25.3, 32.2)	27.5	9.6	48.2
Non-paretic limb						
Stance Time (s)	1.24	0.40	(1.12, 1.35)	1.19	0.69	2.10
Stance %	0.74	0.07	(0.72, 0.76)	0.74	0.58	0.89
Step Length (m)	0.27	0.11	(0.24, 0.30)	0.27	0.09	0.59
Stride Length (m)	0.62	0.24	(0.55, 0.69)	0.59	0.20	1.22
Hip FE ROM	33.5	5.7	(31.8, 35.1)	33.6	18.2	45.0
Knee FE ROM	40.2	7.6	(37.9, 42.4)	39.4	26.6	60.6

$\bar{x}$, mean; s, standard deviation; 95% C.I, 95% confidence interval; Med, median; min, minimum value; max, maximum value; Hip FE ROM, hip flexion/extension range of motion; Knee FE ROM, knee flexion/extension range of motion.

**Table 2 jcm-09-00926-t002:** SI for six gait parameters.

SI for Spatiotemporal and Kinematic Parameters	$\bar{x}$	Med	*s*	Min	Max	*A*
Stance Time (s)	11.7	10.3	11.0	0.0	55.0	1.78
Stance %	11.9	10.6	10.1	0.3	46.9	1.33
Step Length (m)	32.8	23.6	28.9	0.8	107.5	0.85
Stride Length (m)	9.7	6.1	10.2	0.4	48.8	1.80
Hip FE ROM	25.1	16.3	22.0	0.3	84.9	1.23
Knee FE ROM	40.2	30.6	31.7	1.0	114.3	0.75

SI, symmetry index; $\bar{x}$, mean; Med, median; s, standard deviation; min, minimum value; max, maximum value; *A*, skewness coefficient; Hip FE ROM, hip flexion/extension range of motion; Knee FE ROM, knee flexion/extension range of motion.

**Table 3 jcm-09-00926-t003:** Correlations of 3D symmetry indexes to the specific components of and total score in the Wisconsin Gait Scale (WGS).

WGS (items)	Symmetry Indexes Identified for 3D Gait Parameters					
	Stance Time (s)	Stance %	Step Length (m)	Stride Length (m)	Hip FE ROM	Knee FE ROM
**Stance Phase Affected Leg**						
Use of hand-held gait aid	0.12	0.22	0.00	0.13	0.15	0.37 **
Stance time on impaired side	0.80 ***	0.91 ***	0.14	0.43 **	0.43 **	0.49 ***
Step length of unaffected side	0.23	0.36 *	0.10	0.55 ***	0.57 ***	0.51 ***
Weight shift to the affected side	−0.16	−0.21	0.00	0.15	0.15	0.20
Stance width	0.04	0.21	0.11	0.25	0.00	0.20
**Toe Off Affected Leg**						
Guardedness (pause prior to advancing affected leg)	0.24	0.38 **	0.36 *	0.32 *	0.38 **	0.47 ***
Hip extension of affected side	0.44 **	0.50 ***	0.03	0.42 **	0.90 ***	0.63 ***
**Swing Phase Affected Leg**						
External rotation during initial swing	0.31 *	0.26	0.06	0.12	0.24	0.50 ***
Circumduction at mid swing	0.36 *	0.40 **	0.20	0.11	0.25	0.48 ***
Hip hiking at mid swing	0.23	0.36 *	0.20	0.36 *	0.74 ***	0.64 ***
Knee flexion from toe off to mid swing	0.46 **	0.54 ***	0.37 **	0.47 ***	0.57 ***	0.94 ***
Toe clearance	0.23	0.37 **	0.26	0.18	0.34 *	0.34 *
Pelvic rotation at terminal swing	0.13	0.25	0.31 *	0.19	0.27	0.58 ***
**Heel Strike Affected Leg**						
Initial foot contact	0.29 *	0.39 **	0.40 **	0.29 *	0.51 ***	0.60 ***
Total score	0.43 **	0.57 ***	0.29 *	0.43 **	0.65 ***	0.82 ***

3D, 3-dimensional; Hip FE ROM, hip flexion/extension range of motion; Knee FE ROM, knee flexion/extension range of motion; * *p* < 0.05; ** *p* < 0.01; *** *p* < 0.001.

**Table 4 jcm-09-00926-t004:** Presentation of the regression models describing 3-dimensional (3D) Symmetry Indexes with the use of WGS scores.

Dependent Variable	*R* ^2^	Regression Model Formula
Stance Time (s) SI	65.4%	−8.84 + 12.65·WGS_2_
Stance % SI	82.7%	−9.27 + 13.04·WGS_2_
Step Length (m) SI	23.7%	13.58 − 12.58·WGS_7_ + 22.05·WGS_14_
Stride Length (m) SI	46.4%	−15.55 + 5.74·WGS_2_ + 6.66·WGS_3_ + 4.19·WGS_4_
Hip FE ROM SI	87.5%	−20.22 + 22.10·WGS_7_ − 4.37·WGS_9_ + 8.62·WGS_10_
Knee FE ROM SI	88.6%	−40.48+8.87·WGS_1_ + 10.22·WGS_7_ + 28.67·WGS_11_ − 6.01·WGS_14_

*R*^2-^ coefficient of determination, assuming values in the range from 0 to 100% (where 0% reflects a lack of any relationships between the independent variables and the values of the dependent variable, and 100% shows close association between them. The latter extremely rarely encountered in practice). Hip FE ROM, hip flexion/extension range of motion; Knee FE ROM, knee flexion/extension range of motion; WGS_2,14,4,10_, items number 2,14,4,10 in WGS.

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
