# Peer review of "Can an Observational Gait Scale Produce a Result Consistent with Symmetry Indexes Obtained from 3-Dimensional Gait Analysis?: A Concurrent Validity Study"

_jcm, 2020, doi:10.3390/jcm9040926_

Round 1
Reviewer 1 Report
This study attempts to demonstrate that the use of the observational tool Wisconsin Gait Scale (WGS) is substitutable for 3D analysis in a gait lab in post-stroke patients; if true this would potentially result in the potential for a substantial savings in time (as gait analysis is labour-intensive) as well as allowing for savings in costs. The authors tested 50 patients at a chronic stage post-stroke, and compared WGS outcomes to several parameters in terms of Symmetry Indices (SI).
While the use of SI seems a potentially useful compromise in order to normalise gait performance of the affected leg (given the different sizes and strengths of patients -- see lines 44-46), 3D gait analysis does not normally limit itself to SI. Quite the opposite, the best composite index of gait disability is velocity, and the key aspects of gait that can reflect recovery will be coordination metrics such as (Remelius, J. G., Hamill, J., Kent-Braun, J., & Van Emmerik, R. E. (2008). Gait initiation in multiple sclerosis. Motor Control, 12(2), 93-108.). The fundamental problem is that the two legs do not act independently; that is, the non-paretic leg compensates for the paretic leg by diminishing effort. While the Symmetry indices have high specificity for disability, they may have less sensitivity for disability. For this reason, the absolute values of gait parameters are valuable, above and beyond the Symmetry indices. Without an age-matched healthy control group tested in the same laboratory conditions (with the same 3D gait equipment), it is difficult to be certain what disabilities this research fails to detect. For example, Phan et al 2013 found that their stroke patients had a mean step lengths between paretic and nonparetic leg of 50.5 vs 52.7 cm, while the age-matched healthy legs had a step length of 75.5 cm.
Phan, P. L., Blennerhassett, J. M., Lythgo, N., Dite, W., & Morris, M. E. (2013). Over-ground walking on level and sloped surfaces in people with stroke compared to healthy matched adults. Disability and rehabilitation, 35(15), 1302-1307.
The method that the authors used to validate their hypothesis (correlation of the two methods WGS and 3D gait analysis) requires that the data is evenly spaced. As can be seen from the histograms in figure 1, the patient cohort mostly have small changes in SI, while a small group have large changes. This distribution of data will lead to potential false positives where the correlation is fitted well between the cluster of low SI changes and the few extreme SI changes. It is most obvious from table 1, which shows that the parameter step length, which has the most evenly distributed SIs in figure 1, has the lowest correlation to the WGS.
While the authors cannot realistically perform the study again, I think it is essential that the authors add language to the abstract and the discussion highlighting the importance of this limitation. For example,
in lines 247-8: change "therefore the scale can be recommended as a substitute tool in gait assessment" --> "therefore, the scale can be used as a complementary tool in gait assessment."
lines 28-29: change "therefore the scale can be recommended as a substitute tool in gait assessment" --> "therefore, the scale can be used as a complementary tool in gait assessment."
Add in line 210, after the word "problem.": the absolute values of gait parameters, particularly step length, are valuable, above and beyond the Symmetry indices. Without an age-matched healthy control group tested in the same laboratory conditions (with the same 3D gait equipment), it is difficult to be certain what disabilities this research fails to detect.
DETAILS
Table 1 and Table 2 Headings : please change (median) label of "Me" --> "Med"
line 113: add the word "one": "The WGS system allows ONE to asess 14 observable..."
line 127: add the word "of" and the word "a". Remove the words "the fact": "regardless THE FACT whether higher result" --> "regardless OF whether A higher result"
Author Response
Dear Reviewer,
We thank you for reviewing our article titled, “Can an observational gait scale produce a result consistent with symmetry indexes obtained from 3-dimensional gait analysis?: a concurrent validity study”. We have made every effort to improve our manuscript, as guided by the reviewer’s helpful suggestions.
We thank the reviewer of all the comments. Answers are summarised below. All changes are highlighted as red text in the manuscript.
We hope you will be pleased with the changes, and support the publication of our revised manuscript.
With kind regards,
The authors of the article
Point 1: This study attempts to demonstrate that the use of the observational tool Wisconsin Gait Scale (WGS) is substitutable for 3D analysis in a gait lab in post-stroke patients; if true this would potentially result in the potential for a substantial savings in time (as gait analysis is labour-intensive) as well as allowing for savings in costs. The authors tested 50 patients at a chronic stage post-stroke, and compared WGS outcomes to several parameters in terms of Symmetry Indices (SI).
While the use of SI seems a potentially useful compromise in order to normalise gait performance of the affected leg (given the different sizes and strengths of patients -- see lines 44-46), 3D gait analysis does not normally limit itself to SI. Quite the opposite, the best composite index of gait disability is velocity, and the key aspects of gait that can reflect recovery will be coordination metrics such as (Remelius, J. G., Hamill, J., Kent-Braun, J., & Van Emmerik, R. E. (2008). Gait initiation in multiple sclerosis. Motor Control, 12(2), 93-108.). The fundamental problem is that the two legs do not act independently; that is, the non-paretic leg compensates for the paretic leg by diminishing effort. While the Symmetry indices have high specificity for disability, they may have less sensitivity for disability. For this reason, the absolute values of gait parameters are valuable, above and beyond the Symmetry indices. Without an age-matched healthy control group tested in the same laboratory conditions (with the same 3D gait equipment), it is difficult to be certain what disabilities this research fails to detect. For example, Phan et al 2013 found that their stroke patients had a mean step length between paretic and nonparetic leg of 50.5 vs 52.7 cm, while the age-matched healthy legs had a step length of 75.5 cm.
Phan, P. L., Blennerhassett, J. M., Lythgo, N., Dite, W., & Morris, M. E. (2013). Over-ground walking on level and sloped surfaces in people with stroke compared to healthy matched adults. Disability and rehabilitation, 35(15), 1302-1307.
The method that the authors used to validate their hypothesis (correlation of the two methods WGS and 3D gait analysis) requires that the data is evenly spaced. As can be seen from the histograms in figure 1, the patient cohort mostly have small changes in SI, while a small group have large changes. This distribution of data will lead to potential false positives where the correlation is fitted well between the cluster of low SI changes and the few extreme SI changes. It is most obvious from table 1, which shows that the parameter step length, which has the most evenly distributed SIs in figure 1, has the lowest correlation to the WGS.
While the authors cannot realistically perform the study again, I think it is essential that the authors add language to the abstract and the discussion highlighting the importance of this limitation. For example,
in lines 247-8: change "therefore the scale can be recommended as a substitute tool in gait assessment" --> "therefore, the scale can be used as a complementary tool in gait assessment."
lines 28-29: change "therefore the scale can be recommended as a substitute tool in gait assessment" --> "therefore, the scale can be used as a complementary tool in gait assessment."
Add in line 210, after the word "problem.": the absolute values of gait parameters, particularly step length, are valuable, above and beyond the Symmetry indices. Without an age-matched healthy control group tested in the same laboratory conditions (with the same 3D gait equipment), it is difficult to be certain what disabilities this research fails to detect.
Response 1: Thank you for this valuable comment. We fully agree with the Reviewer. Gait velocity is a powerful indicator of function and prognosis after stroke. A motivation of the current study was provided by the comment by Wellmon et al., The WGS provides a valuable and complementary understanding of walking performance that is not possible by only examining gait speed. Reductions in temporal and spatial gait asymmetries is associated with recovery and increased walking stability after a stroke. In fact, this highly relevant Reviewer comment provides an excellent rationale for further research.
In accordance with the Reviewer’s suggestion, we have added language to the abstract and the discussion highlighting the importance of the relevant limitation.
BEFORE
lines 247-8:"therefore the scale can be recommended as a substitute tool in gait assessment"
lines 28-29:"therefore the scale can be recommended as a substitute tool in gait assessment"
lines 208-210: This suggests that spatiotemporal and kinematic gait pattern asymmetry in individuals after stroke is a serious problem.
AFTER
lines 329-330: ", therefore, the scale can be used as a complementary tool in gait assessment."
lines 27-28: "therefore, the scale can be used as a complementary tool in gait assessment."
Lines 277-282: This suggests that spatiotemporal and kinematic gait pattern asymmetry in patients after stroke is a serious problem: the absolute values of gait parameters, particularly step length, are valuable, above and beyond the Symmetry indices. Without an age-matched healthy control group tested in the same laboratory conditions (with the same 3D gait equipment), it is difficult to be certain what disabilities this research fails to detect.
Point 2:
DETAILS
Table 1 and Table 2 Headings : please change (median) label of "Me" --> "Med"
line 113: add the word "one": "The WGS system allows ONE to asess 14 observable..."
line 127: add the word "of" and the word "a". Remove the words "the fact": "regardless THE FACT whether higher result" --> "regardless OF whether A higher result"
Response 2: Thank you for the helpful suggestions, all the comments have been taken into account.
BEFORE
Table 1 and Table 2 "Me"
line 113: " The WGS allows to assess 14 observable"
line 127: "regardless the fact whether higher result"
AFTER
Table 1 and Table 2 "Med "
line 165: " The WGS system allows one to assess 14 observable"
line 192: " regardless of whether a higher result "

Reviewer 2 Report
Manuscript title: Can an observational gait scale produce a result consistent with symmetry indexes obtained from 3-dimensional gait analysis: a concurrent validity study
General Comments
This paper attempts to validate observational gait analysis to more traditional 3-dimensional motion analysis. While this is an interesting though, the only focus of the project is gait symmetry. I feel this truly oversimplifies gait analyses, as gait symmetry is only a small component of gait function and post-stroke function. I highly encourage the authors to review the information they have presented and the arguments they have made.
Introduction
The main problem associated with the introduction is when the authors state “As a result, the walking pattern is described mainly in terms of gait symmetry. Therefore in the current study, the authors decided to apply symmetry indexes”. This statement is misleading as SI is only one method of analyzing gait. Further, simply knowing that an individual exhibits asymmetrical gait is not as helpful as knowing how much asymmetry there is in the system. I think the authors need to critically review this concept and provide greater detail on this. So what if an individual can observe symmetry? There is little relevance in the binary answer of yes or no in clinical populations and gait symmetry.
On page 2, line 50, the authors mention conventional methods of gait analysis but do not provide any context. Again, in the context of symmetry or SI, there is very little information that is helpful when just determining whether an individual exhibits asymmetrical gait, and even more so with basic spatio-temporal asymmetry.
Methods
I have used the Robinson index previously, but my main concern with it is that there isn’t a value where we understand what is bad, nor what is acceptable. The authors should consider providing greater information how this method is useful.
The authors need to specify whether the Physical Therapist who reviewed the gait assessments did so via video or by examining the data.
Discussion
I understand the purpose of the study however, my main concern is the oversimplification of clinical gait analysis and the lack of mentioning of other tools besides traditional 3-dimensional motion capture. Tools such as a GaitRite, Inertial Measurement Units, to name are few, are validated tools which clinicians can use to aid their analysis. Simply observing symmetry may not provide a comprehensive picture of gait function. I believe the authors truly need to critically review the overall purpose and potential pitfalls of this study.
Author Response
Dear Reviewer,
We thank you for reviewing our article titled, “Can an observational gait scale produce a result consistent with symmetry indexes obtained from 3-dimensional gait analysis?: a concurrent validity study”. We have made every effort to improve our manuscript, as guided by the reviewer’s helpful suggestions.
We thank the reviewer of all the comments. Answers are summarised below. All changes are highlighted as red text in the manuscript.
We hope you will be pleased with the changes, and support the publication of our revised manuscript.
With kind regards,
The authors of the article
Point 1: General Comments
This paper attempts to validate observational gait analysis to more traditional 3-dimensional motion analysis. While this is an interesting though, the only focus of the project is gait symmetry. I feel this truly oversimplifies gait analyses, as gait symmetry is only a small component of gait function and post-stroke function. I highly encourage the authors to review the information they have presented and the arguments they have made.
Introduction
The main problem associated with the introduction is when the authors state “As a result, the walking pattern is described mainly in terms of gait symmetry. Therefore in the current study, the authors decided to apply symmetry indexes”. This statement is misleading as SI is only one method of analyzing gait. Further, simply knowing that an individual exhibits asymmetrical gait is not as helpful as knowing how much asymmetry there is in the system. I think the authors need to critically review this concept and provide greater detail on this. So what if an individual can observe symmetry? There is little relevance in the binary answer of yes or no in clinical populations and gait symmetry.
On page 2, line 50, the authors mention conventional methods of gait analysis but do not provide any context. Again, in the context of symmetry or SI, there is very little information that is helpful when just determining whether an individual exhibits asymmetrical gait, and even more so with basic spatio-temporal asymmetry.
Response 1: Thank you for this valuable comment. The motivation for our research was provided by the studies published by Patterson et al. (2008; 2010) who emphasised that nowadays more and more attention is paid to symmetry, as a basic gait characteristic, particularly in patients after stroke. These researchers also pointed out that, from clinical viewpoint, gait symmetry is of essential importance as it may be linked to various negative consequences, including inefficiency, poor balance control, loss of bone mass density in the paretic lower limb as well as increased risk of musculoskeletal injury to the unaffected lower limb [Patterson et al. 2010]. Therefore, it is important to identify and understand any existing asymmetries, found in various gait parameters. In line with this requirement, in the WGS higher scores are indicative of greater gait asymmetry [Rodriquez et al., Pizzi et al., Wellmon et al.]. A number of inter-limb deviations account for symmetry or asymmetry, which is a significant feature of gait. Importantly, in the related literature the terms symmetry and asymmetry are commonly used when referring to the same parameters [Viteckova et al.]. Healthy individual generally present symmetrical gait, with only minor deviations. Hence it seems that normal and pathological gait patterns are effectively differentiated by the characteristic of asymmetry, or lack of symmetry [Patterson et al. 2008].
Another reason for selecting this specific subject matter for our research lies in the fact that many studies have already reported findings on the feasibility of the WGS as a tool for observational assessment of gait asymmetry in patients post-stroke [Pizzi et al., Lu et al., Rodriquez et al., Yaliman et al., Wellmon et al.]. The specific items of the scale focus on positions assumed by parts of the lower extremities and joints during the gait cycle, taking into account both the affected and the unaffected leg, which are then compared.
The WGS makes it possible to identify basic defects in the gait pattern, such as asymmetries observed in step length, duration of gait phases, and body weight shifting, as well as impaired mobility in the joints of the paretic leg. As a result, the WGS provides qualitative description of gait mainly in terms of gait asymmetry [Rodriquez et al., Pizzi et al., Wellmon et al.]. According to Pizzi et al. and Lu et al. the observational WGS measures changes occurring in hemiplegic gait, and reflected by the inter-limb symmetry during movement, weight shift and weight-bearing during the stance and swing phases as well as the ankle, knee and hip kinematics, balance/guardedness, and the use of assistive devices. Wellmon et al. argued that the WGS provides important complementary information on patients’ walking performance, making it possible to identify reductions in spatial and temporal gait asymmetries which are indicative of recovery and better walking stability in patients with stroke.
In the case of patients with hemiplegia, asymmetries are observed in stride time and length as well as the timing in the single-limb support phase on the paretic and non-paretic limbs [Bohannon et al., Goldie et al., Wall and Turnbull]. Furthermore, temporal and spatial asymmetries are commonly found in a number of gait measures in comparative assessments of the hemiparetic and the unaffected side [Patterson et al.]. There may be inequalities in step length resulting from efforts aimed at reducing weight bearing on the paretic limb. It is also possible to observe shorter stance time on the paretic limb [von Schroeder et al.], as well as increased base of support during double stance and difficulty clearing the paretic foot during the swing phase [Goldie et al.]. Other researchers have reported differences in trunk alignment and limited hip and knee flexion and extension [Bensoussan et al., Boudarham et al.]. Collectively, these changes lead to asymmetrical gait patterns. Given this, the aim of a gait rehabilitation program in patients with stroke is generally to improve gait symmetry [Friedman]. The WGS enables observational assessment of gait temporal symmetry, spatial symmetry and kinematic symmetry [Pizzi et al., Lu et al., Rodriquez et al.]. These observations provided impulse for this study which was designed to determine whether, and to what extent, observational information about gait symmetry obtained from WGS-based assessment can be used as a complementary tool and can be applied to predict results of 3D gait analysis for selected symmetry indicators related to spatiotemporal and kinematic gait parameters.
In accordance with the Reviewer’s suggestion, we have provided the context.
REFERENCES:
- Patterson, K.K.; Parafianowicz, I.; Danells, C.J.; Closson, V.; Verrier, M.C.; Staines, W.R.; Black, S.E.; McIlroy, W.E. Gait asymmetry in community-ambulating stroke survivors. Phys. Med. Rehabil. 2008, 89, 304-310.
- Patterson, K.K.; Gage, W.H.; Brooks, D.; Black, S.E.; McIlroy, W.E. Evaluation of gait symmetry after stroke: a comparison of current methods and recommendations for standardization. Gait Posture. 2010, 31, 241-246.
- Rodriquez, A.A.; Black, P.O.; Kile, K.A.; Sherman, J.; Stellberg, B.; McCormick, J.; Roszkowski, J.; Swiggum, E. Gait training efficacy using a home-based practice model in chronic hemiplegia. Phys. Med. Rehabil. 1996, 77, 801–805.
- Pizzi, A.; Carlucci, G.; Falsini, C.; Lunghi, F.; Verdesca, S.; Grippo, A. Gait in hemiplegia: Evaluation of clinical features with the Wisconsin Gait Scale. Rehabil. Med. 2007, 39, 170-174.
- Wellmon, R.; Degano, A.; Rubertone, J.A.; Campbell, S.; Russo, K.A. Interrater and intrarater reliability and minimal detectable change of the Wisconsin Gait Scale when used to examine videotaped gait in individuals post-stroke. Physiother. 2015, 5, 11.
- Viteckova, S.; Kutilek, P.; Svoboda, Z,; Krupicka, R.; Kauler, J.; Szabo, Z. Gait symmetry measures: A review of current and prospective methods. Signal. Proces. 2018, 42, 89-100.
- Lu, X.; Hu, N.; Deng, S.; Li, J.; Qi, S.; Bi, S. The reliability, validity and correlation of two observational gait scales assessed by video tape for Chinese subjects with hemiplegia. J. Phys. Ther. Sci. 2015, 27, 3717-3721.
- Yaliman, A.; Kesiktas, N.; Ozkaya, M.; Eskiyurt, N.; Erkan, O.; Yilmaz, E. Evaluation of intrarater and interrater reliability of the Wisconsin Gait Scale with using the video taped stroke patients in a Turkish sample. 2014, 34, 253-258.
- Bohannon, R.W.; Horton, M.G.; Wikholm, J.B. Importance of four variables of walking to patients with stroke. J. Rehab. Res. 1991, 14, 246–250.
- Goldie, P.A.; Matyas, T.A.; Evans, O.M. Gait after stroke: initial deficit and changes in temporal patterns for each gait phase. Phys. Med. Rehabil. 2001, 82, 1057–1065.
- Wall, J.; Turnbull, G. Gait asymmetries in residual hemiplegia. Phys. Med. Rehabil. 1986, 67, 550–553.
- von Schroeder, H.P.; Coutts, R.D.; Lyden, P.D.; Billings, E.Jr.; Nickel, V.L. Gait parameters following stroke: a practical assessment. Rehabil. Res. Dev. 1995, 32, 25-31.
- Bensoussan, L.; Mesure, S.; Viton, J.M.; Delarque, A. Kinematic and kinetic asymmetries in hemiplegic patients' gait initiation patterns. Rehabil. Med. 2006, 38, 287-294.
- Boudarham, J.; Roche, N.; Pradon, D.; Bonnyaud, C.; Bensmail, D.; Zory, R. Variations in Kinematics during Clinical Gait Analysis in Stroke Patients. PLoS One. 2013, 8, e66421.
- Friedman, P.J. Gait recovery after hemiplegic stroke. Disabil. Stud. 1990, 12, 119–122.
BEFORE
lines 38-61: The WGS consists of four subscales, and assesses 14 observable gait parameters occurring during the consecutive gait phases, i.e. stance phase, toe off, swing phase and heel strike of the affected leg. The WGS has been shown to be accurate and reliable, therefore it can effectively be used to evaluate progress in gait rehabilitation after stroke [3-8]. The specific items of the scale focus on positions assumed by parts of the lower extremities and joints during the gait cycle, taking into account both the affected and the unaffected leg, which are then compared. As a result, the walking pattern is described mainly in terms of gait symmetry [5,8]. Therefore, in the current study the authors decided to apply Symmetry Indexes (SI), calculated based on 3D gait parameters, and to compare the results of equipment-based 3D assessment to those obtained using observational gait analysis performed using the subjective WGS.
Since it reflects similarities in spatiotemporal and kinematic parameters of the right and left lower limb, symmetry is a significant measure of gait assessment, and can more effectively describe post-stroke gait mechanisms, compared to conventional methods. Quality of gait, reflected by symmetry in step length and duration of gait phases, tends to change towards greater asymmetry at later stages post stroke [9-11]. Additionally, Patterson et al. [12] argue that stance time, step length and swing time are the most important spatiotemporal parameters of gait symmetry post-stroke. On the other hand, Boudarham et al. [13] reported that predictors of walking performance in hemiplegic patients include hip impairment and inadequate knee function, affecting kinematic gait parameters. The WGS enables observational assessment of gait symmetry for such factors as stance time (temporal symmetry), step length (spatial symmetry), hip and knee range of motion (kinematic symmetry) [5,8].
The aim of this study was to determine whether, and to what extent, observational information obtained from WGS-based assessment can be applied to predict results of 3D gait analysis for selected symmetry indicators related to spatiotemporal and kinematic gait parameters.
AFTER
lines 38-109: The Wisconsin Gait Scale (WGS), which is an observational tool enabling assessment of gait quality in patients after stroke, was developed by Rodriguez et al. [3]. The WGS consists of four subscales, and assesses 14 observable gait parameters occurring during the consecutive gait phases, i.e. stance phase, toe off, swing phase and heel strike of the affected leg. The WGS has been shown to be accurate and reliable in patients post-stroke, therefore it can effectively be used to evaluate progress in gait rehabilitation after stroke [3-8]. Originally, the WGS was intended as a tool enabling assessment of effects achieved during a home-based gait training programme designed for patients at a chronic stage of recovery post-stroke [3]. The authors subsequently tested the validity of the WGS, and showed its effectiveness as an instrument of gait measurement making it possible to compare outcomes [3]. Similarly, Yaliman et al. [7] and Wellmon et al. [8] established that the WGS is highly reliable, and they suggested it may be used as an objective tool for recording results of an observational gait analysis after stroke. Estrada-Barranco et al. [4] showed that the correlation of the WGS with the balance and functionality scales in stroke patients at subacute and chronic stages was excellent. Pizzi et al. [6] reported that the WGS may effectively be used to evaluate qualitative changes in gait patterns presented by patients with post-stroke hemiparesis, and to assess changes observed in course of long-term rehabilitation. The tool may effectively be applied when it is necessary to perform targeted and standardised measurements of hemiplegic gait in order to accurately modify rehabilitation in line with the results of monitoring [6].
The respective parts of the WGS focus on spatiotemporal (subscale one) and kinematic parameters of gait (subscale one, two, three and four). The video recording is analysed by the examiner, and based on that a score is assigned for the performance of the gait pattern [3]. The specific items of the scale focus on positions assumed by parts of the lower extremities and joints during the gait cycle, taking into account both the affected and the unaffected leg, which are then compared. The WGS makes it possible to identify basic defects in the gait pattern, such as asymmetries observed in step length, duration of gait phases, and body weight shifting, as well as impaired mobility in the joints of the paretic leg. As a result, the WGS provides qualitative description of gait mainly in terms of gait asymmetry [3,6,8]. According to Pizzi et al. [6] and Lu et al. [9] the observational WGS measures changes occurring in hemiplegic gait, and reflected by the inter-limb symmetry during movement, weight shift and weight-bearing during the stance and swing phases as well as the ankle, knee and hip kinematics, balance/guardedness, and the use of assistive devices. Wellmon et al. [8] argued that the WGS provides important complementary information on patients’ walking performance, making it possible to identify reductions in spatial and temporal gait asymmetries which are indicative of recovery and better walking stability in patients with stroke.
The motivation for our research was provided by the studies published by Patterson et al. [10,11] who emphasised that nowadays more and more attention is paid to symmetry, as a basic gait characteristic, particularly in patients after stroke. These researchers also pointed out that, from clinical viewpoint, gait symmetry is of essential importance as it may be linked to various negative consequences, including inefficiency, poor balance control, loss of bone mass density in the paretic lower limb as well as increased risk of musculoskeletal injury to the unaffected lower limb [11]. Therefore, it is important to identify and understand any existing asymmetries, found in various gait parameters. In line with this requirement, in the WGS higher scores are indicative of greater gait asymmetry [3,6,8]. A number of inter-limb deviations account for symmetry or asymmetry, which is a significant feature of gait. Importantly, in the related literature the terms symmetry and asymmetry are commonly used when referring to the same parameters [12]. Healthy individual generally present symmetrical gait, with only minor deviations. Hence it seems that normal and pathological gait patterns are effectively differentiated by the characteristic of asymmetry, or lack of symmetry [10]. Another reason for selecting this specific subject matter for our research lies in the fact that many studies have already reported findings on the feasibility of the WGS as a tool for observational assessment of gait asymmetry in patients post-stroke [3,6-8].
Since it reflects similarities in spatiotemporal and kinematic parameters of the right and left lower limb, symmetry is a significant measure of gait assessment, and can more effectively describe post-stroke gait mechanisms, compared to conventional measures, such as velocity. Furthermore, it may provide important information for clinicians, to be taken into account in treatment decisions [13]. In the case of patients with hemiplegia, asymmetries are observed in stride time and length as well as the timing in the single-limb support phase on the paretic and non-paretic limbs [14-16]. Furthermore, temporal and spatial asymmetries are commonly found in a number of gait measures in comparative assessments of the hemiparetic and the unaffected side [11]. There may be inequalities in step length resulting from efforts aimed at reducing weight bearing on the paretic limb. It is also possible to observe shorter stance time on the paretic limb [17], as well as increased base of support during double stance and difficulty clearing the paretic foot during the swing phase [15]. Other researchers have reported differences in trunk alignment and limited hip and knee flexion and extension [18,19]. Quality of gait, reflected by symmetry in step length and duration of gait phases, tends to change towards greater asymmetry at later stages post stroke [13]. Given this, the aim of a gait rehabilitation program in patients with stroke is generally to improve gait symmetry [20]. The WGS enables observational assessment of gait symmetry for such factors as stance time (temporal symmetry), step length (spatial symmetry), hip and knee range of motion (kinematic symmetry) [3,8].
The aim of this study was to determine whether, and to what extent, observational information about gait symmetry obtained from WGS-based assessment can be used as a complementary tool and can be applied to predict results of 3D gait analysis for selected symmetry indicators related to spatiotemporal and kinematic gait parameters.
Point 2: Methods
I have used the Robinson index previously, but my main concern with it is that there isn’t a value where we understand what is bad, nor what is acceptable. The authors should consider providing greater information how this method is useful.
The authors need to specify whether the Physical Therapist who reviewed the gait assessments did so via video or by examining the data.
Response 2: Thank you for helpful suggestions. We have provided more information about the method, which is used by numerous researchers assessing patients after stroke [Patterson et al., Kim et al., Titianova et al., Bovonsunthonchai et al.], and other cohorts. This index has been used to analyse symmetry in long distance runners, and in healthy subjects, as well as leg length discrepancy, and amputees [Sadeghi et al]. For the ideal symmetrical gait condition, the symmetry index should be zero, reflecting perfectly symmetrical gait pattern. Higher symmetry index values correspond to greater gait asymmetry [Robinson et al., Bovonsunthonchai et al.].
In accordance with the Reviewer’s suggestion, we have specified that Physical Therapist reviewed the gait assessments via video. Based on the analysis of the video, the evaluator assigned scores for the performance of the gait pattern.
REFERENCES:
- Kim, C.M.; Eng, J.J. Symmetry in vertical ground reaction force is accompanied by symmetry in temporal but not distance variables of gait in persons with stroke. Gait Posture. 2003, 18, 23-28.
- Titianova, E.B.; Tarkka, I.M. Asymmetry in walking performance and postural sway in patients with chronic unilateral cerebral infarction Rehabil. Res. Dev. 1995, 32, 236-244.
- Bovonsunthonchai, S.; Hiengkaew, V,; Vachalathiti, R.; Vongsirinavarat, M. Gait symmetrical indexes and their relationships to muscle tone, lower extremity function, and postural balance in mild to moderate stroke. Med. Assoc. Thai. 2011 94, 476-484.
- Sadeghi, H.; Allard, P.; Prince, F.; Labelle, H. Symmetry and limb dominance in able-bodied gait: a review. Gait Posture. 2000, 12, 34–45.
- Robinson, R.O.; Herzog, W.; Nigg, B.M. Use of force platform variables to quantify the effects of chiropractic manipulation on gait symmetry. Manip. Physiol. Ther. 1987, 10, 172-176.
BEFORE
lines 122-133: Measurement of gait symmetry applied the most commonly used method – absolute index proposed by Robinson [16], which is calculated as a quotient of the absolute difference between the measures for both legs and the mean of these measures, multiplied by 100. The absolute value of the difference between the affected and the unaffected side was taken into account because gait defects are reflected by the disparity between the results identified for both legs, regardless of whether a higher result is found for the right or the left leg. Given the method applied in determining the symmetry index (it can only assume positive values), we can expect that its distribution will be concentrated around values approaching 0. Zero value of the above index reflects perfect symmetry [16].
lines 111-113: The recordings and WGS-based gait assessment were reviewed and interpreted by a physical therapist with over 10 years of experience in working with patients post-stroke, and with expertise in using the WGS and interpreting the scores.
AFTER
lines 181-197: The analyses took into account six gait symmetry indexes calculated based on 3D assessment involving 50 patients. In the current study the authors decided to apply Symmetry Indexes (SI), calculated based on 3D gait parameters, and to compare the results of equipment-based 3D assessment to those obtained using observational gait analysis performed using the subjective WGS. The measurement of gait symmetry applied the most commonly used method – absolute index proposed by Robinson [24], and used by numerous researchers assessing patients after stroke [11,25-27], and other cohorts. This index has been used to analyse symmetry in long distance runners, and in healthy subjects, as well as leg length discrepancy, and amputees [28]. It is calculated as a quotient of the absolute difference between the measures for both legs and the mean of these measures, multiplied by 100. The absolute value of the difference between the affected and the unaffected side was taken into account because gait defects are reflected by the disparity between the results identified for both legs, regardless of whether a higher result is found for the right or the left leg. Given the method applied in determining the symmetry index (it can only assume positive values), we can expect that its distribution will be concentrated around values approaching zero. For the ideal symmetrical gait condition, the symmetry index should be zero, reflecting perfectly symmetrical gait pattern. Higher symmetry index values correspond to greater gait asymmetry [24, 27].
lines 161-165: The recordings and WGS-based gait assessment were reviewed and interpreted by a physical therapist with over 10 years of experience in working with patients post-stroke, and with expertise in using the WGS and interpreting the scores. A physical therapist reviewed the gait assessments via video. Based on the analysis of the video recordings, the evaluator assigns scores for the performance of the gait pattern.
Point 3: Discussion
I understand the purpose of the study however, my main concern is the oversimplification of clinical gait analysis and the lack of mentioning of other tools besides traditional 3-dimensional motion capture. Tools such as a GaitRite, Inertial Measurement Units, to name are few, are validated tools which clinicians can use to aid their analysis. Simply observing symmetry may not provide a comprehensive picture of gait function. I believe the authors truly need to critically review the overall purpose and potential pitfalls of this study.
Response 3: Thank you for this valuable comment. We fully agree with the Reviewer. In accordance with the Reviewer’s suggestion, we have added missing information and study limitations. We have supplemented the purpose and the conclusion section. In fact, this highly relevant comment provides an excellent rationale for further research.
BEFORE
lines 235-239: Our study provides evidence for effectiveness of the observational gait analysis based on the WGS, which also enables comprehensive objective assessment of both spatiotemporal and kinematic parameters asymmetry. In situations when equipment-based gait assessment systems are not available, the regression models described here may be helpful for physicians and physiotherapists, enabling fairly accurate estimation of selected 3D symmetry indexes from WGS scores.
lines 59-61: The aim of this study was to determine whether, and to what extent, observational information obtained from WGS-based assessment can be applied to predict results of 3D gait analysis for selected symmetry indicators related to spatiotemporal and kinematic gait parameters
lines 246-248: The study confirms that observation of gait using the WGS, which is an ordinal scale, is consistent with the main aims of 3D assessment, therefore the scale can be recommended as a substitute tool in gait assessment.
AFTER
lines 308-321: Besides the traditional 3-dimensional motion capture, a number of advanced, objective tools for gait analysis are available at present, e.g. GaitRite or Inertial Measurement Units. However, they are all costly and require technical expertise and specialist equipment. The current findings are not intended to undermine the importance of the objective gait assessment methods, which provide gold standard in this area; they only show that the WGS enables accurate assessment of gait, corresponding to the objective 3D gait measures. The findings suggest that in situations when costly equipment-based gait assessment systems are not accessible to medical facilities for various reasons, the regression models described here may be helpful for physicians and physiotherapists, enabling rough estimation of selected 3D symmetry indexes based on WGS scores. The study shows that this simple, easy and time-effective scale can be used as a complementary tool in gait assessment and can be recommended as an alternative tool when 3DGA is unavailable. A limitation of the present study lies in the fact that traditional 3-dimensional motion capture was the only objective gait analysis method applied. Therefore, it is necessary to continue the research taking into account other advanced tools for clinical gait analysis.
lines 106-109: The aim of this study was to determine whether, and to what extent, observational information about gait symmetry obtained from WGS-based assessment can be used as a complementary tool and can be applied to predict results of 3D gait analysis for selected symmetry indicators related to spatiotemporal and kinematic gait parameters.
lines 328-330: The study confirms that observation of gait using the WGS, which is an ordinal scale, is consistent with the selected aims of 3D assessment, therefore, the scale can be used as a complementary tool in gait assessment.

Reviewer 3 Report
This article has good clinical utility and the information would be useful to healthcare professionals who have roles in assessing gait characteristics. There are modifications that could be made in order to improve the clarity of the topic.
Line 38 – The authors introduce the WGS however it would help readers not familiar with this scale understand this tool by expanding this section to provide more detail on what the WGS is and how it is administered.
Line 40 – States that the WGS is accurate and reliable – is this in the stroke population?
Lines 44 – 46 – This statement seems to fit more in the method section and not the introduction.
Lines 50 -52- There seems to be a discrepancy in the citation. The statement refers to gait quality declining in later stages post-stroke but the references cited [9-11] do not reflect this. However, later in the discussion section, this is mentioned again, with the correct citation [22].
Line 76 – the correct term for the cognitive test is the Mini-Mental State Examination. Also, why was a score of 20 selected as the cut-off? Additionally, one exclusion criteria is “mobility deficits significantly limiting and disrupting the patient’s ability to walk”. The participants in this study all have disrupted gait, therefore is this statement intended to mean those that cannot walk independently? May need to be more clear.
Line 92 – what was the distance the participant had to walk in one pass?
In the method, it was described that markers were placed to capture motions at all joints of the lower extremity, including the ankle/foot. The WGS includes items that can be correlated to what it happening at the ankle/foot (e.g. heel strike). Additionally, one reference mentioned in this article [13] reported on the effect of abnormal ankle/foot function on gait symmetry. Why was the ankle/foot not included in this study?
The conclusion reports that the WGS can be a substitute for main aims of 3D assessment, however the results did not support use of any of the WGS components for the assessment of step length and stride length, vital components of the gait cycle.
There are several terms used in this paper to describe the participants of the study (patients, individuals, participants). Recommend modifying for consistent terminology.
Author Response
Dear Reviewer,
We thank you for reviewing our article titled, “Can an observational gait scale produce a result consistent with symmetry indexes obtained from 3-dimensional gait analysis?: a concurrent validity study”. We have made every effort to improve our manuscript, as guided by the reviewer’s helpful suggestions.
We thank the reviewer of all the comments. Answers are summarised below. All changes are highlighted as red text in the manuscript.
We hope you will be pleased with the changes, and support the publication of our revised manuscript.
With kind regards,
The authors of the article
Point 1: This article has good clinical utility and the information would be useful to healthcare professionals who have roles in assessing gait characteristics. There are modifications that could be made in order to improve the clarity of the topic.
Line 38 – The authors introduce the WGS however it would help readers not familiar with this scale understand this tool by expanding this section to provide more detail on what the WGS is and how it is administered.
Response 1: Thank you for the helpful comment. We have supplemented the text with more comprehensive description of the WGS and how it is administered.
BEFORE
lines 38-44: The WGS consists of four subscales, and assesses 14 observable gait parameters occurring during the consecutive gait phases, i.e. stance phase, toe off, swing phase and heel strike of the affected leg. The WGS has been shown to be accurate and reliable, therefore it can effectively be used to evaluate progress in gait rehabilitation after stroke [3-8]. The specific items of the scale focus on positions assumed by parts of the lower extremities and joints during the gait cycle, taking into account both the affected and the unaffected leg, which are then compared. As a result, the walking pattern is described mainly in terms of gait symmetry [5,8].
lines 111-119: The recordings and WGS-based gait assessment were reviewed and interpreted by a physical therapist with over 10 years of experience in working with patients post-stroke, and with expertise in using the WGS and interpreting the scores. The WGS allows to assess 14 observable gait parameters (as described in the Introduction above). The total score, in the range of 13.35 - 42 points, is calculated for all the items. The points assigned to items 2–10 and 12-14 are added up. Responses to items 1 and 11 are weighted by 3/5 and 3/4, respectively and then the points are added to the total score. Higher scores correspond to poorer overall walking performance and more visible gait deviations [5.8]. Good intra- and inter-rater reliability of the WGS was demonstrated by a number of studies [6,8,15-17].
AFTER
lines 38-71: The Wisconsin Gait Scale (WGS), which is an observational tool enabling assessment of gait quality in patients after stroke, was developed by Rodriguez et al. [3]. The WGS consists of four subscales, and assesses 14 observable gait parameters occurring during the consecutive gait phases, i.e. stance phase, toe off, swing phase and heel strike of the affected leg. The WGS has been shown to be accurate and reliable in patients post-stroke, therefore it can effectively be used to evaluate progress in gait rehabilitation after stroke [3-8]. Originally, the WGS was intended as a tool enabling assessment of effects achieved during a home-based gait training programme designed for patients at a chronic stage of recovery post-stroke [3]. The authors subsequently tested the validity of the WGS, and showed its effectiveness as an instrument of gait measurement making it possible to compare outcomes [3]. Similarly, Yaliman et al. [7] and Wellmon et al. [8] established that the WGS is highly reliable, and they suggested it may be used as an objective tool for recording results of an observational gait analysis after stroke. Estrada-Barranco et al. [4] showed that the correlation of the WGS with the balance and functionality scales in stroke patients at subacute and chronic stages was excellent. Pizzi et al. [6] reported that the WGS may effectively be used to evaluate qualitative changes in gait patterns presented by patients with post-stroke hemiparesis, and to assess changes observed in course of long-term rehabilitation. The tool may effectively be applied when it is necessary to perform targeted and standardised measurements of hemiplegic gait in order to accurately modify rehabilitation in line with the results of monitoring [6].
The respective parts of the WGS focus on spatiotemporal (subscale one) and kinematic parameters of gait (subscale one, two, three and four). The video recording is analysed by the examiner, and based on that a score is assigned for the performance of the gait pattern [3]. The specific items of the scale focus on positions assumed by parts of the lower extremities and joints during the gait cycle, taking into account both the affected and the unaffected leg, which are then compared. The WGS makes it possible to identify basic defects in the gait pattern, such as asymmetries observed in step length, duration of gait phases, and body weight shifting, as well as impaired mobility in the joints of the paretic leg. As a result, the WGS provides qualitative description of gait mainly in terms of gait asymmetry [3,6,8]. According to Pizzi et al. [6] and Lu et al. [9] the observational WGS measures changes occurring in hemiplegic gait, and reflected by the inter-limb symmetry during movement, weight shift and weight-bearing during the stance and swing phases as well as the ankle, knee and hip kinematics, balance/guardedness, and the use of assistive devices. Wellmon et al. [8] argued that the WGS provides important complementary information on patients’ walking performance, making it possible to identify reductions in spatial and temporal gait asymmetries which are indicative of recovery and better walking stability in patients with stroke.
lines 161-179: The recordings and WGS-based gait assessment were reviewed and interpreted by a physical therapist with over 10 years of experience in working with patients post-stroke, and with expertise in using the WGS and interpreting the scores. A physical therapist reviewed the gait assessments via video. Based on the analysis of the video recordings, the evaluator assigns scores for the performance of the gait pattern. The WGS system allows one to assess 14 observable gait parameters. The five initial items relate to affected leg stance phase, i.e. 1- use of hand-held assistive device, 2- stance time on the affected side, 3- step length on the unaffected side, 4- weight shift to the paretic limb with/without assistive device, and 5- stance width. The next part of the scale relates to the affected leg toe-off phase and comprises two items: 6-guardedness (pause before advancing the paretic leg), 7- affected leg hip extension (observation of gluteal crease). The third subscale covers the affected leg swing phase, and comprises six items: 8- external rotation during initial swing, 9- circumduction at mid-swing (path of the paretic foot), 10- hip hiking at mid-swing, 11- knee flexion from toe off to mid-swing, 12- toe clearance, 13- pelvic rotation at terminal swing. The final subscale relates to the affected leg heel strike and comprises only one item: 14- initial foot contact of the affected leg. The total score, in the range of 13.35 - 42 points, is calculated for all the items. The points assigned to items 2–10 and 12-14 are added up. Responses to items 1 and 11 are weighted by 3/5 and 3/4, respectively and then the points are added to the total score. Higher scores correspond to poorer overall walking performance and more visible gait deviations [3.8]. Good intra- and inter-rater reliability of the WGS in patients post-stroke was demonstrated by a number of studies [6-9,22,23].
Point 2: Line 40 – States that the WGS is accurate and reliable – is this in the stroke population?
Response 2: Thank you for this comment, yes this relates to the stroke population.
BEFORE
line 40: The WGS has been shown to be accurate and reliable
AFTER
line 42: The WGS has been shown to be accurate and reliable in patients post-stroke
Point 3: Lines 44 – 46 – This statement seems to fit more in the method section and not the introduction.
Response 3: Thank you for this comment. In accordance with the Reviewer’s suggestion, we have moved the statement to the method section.
BEFORE
lines 44-46: Therefore, in the current study the authors decided to apply Symmetry Indexes (SI), calculated based on 3D gait parameters, and to compare the results of equipment-based 3D assessment to those obtained using observational gait analysis performed using the subjective WGS.
AFTER
lines 181-184: The analyses took into account six gait symmetry indexes calculated based on 3D assessment involving 50 patients. In the current study the authors decided to apply Symmetry Indexes (SI), calculated based on 3D gait parameters, and to compare the results of equipment-based 3D assessment to those obtained using observational gait analysis performed using the subjective WGS.
Point 4: Lines 50 -52- There seems to be a discrepancy in the citation. The statement refers to gait quality declining in later stages post-stroke but the references cited [9-11] do not reflect this. However, later in the discussion section, this is mentioned again, with the correct citation [22].
Response 4: Thank you for the helpful suggestion. We apologise for this mistake. In accordance with the Reviewer’s suggestion, we have used the correct citation.
BEFORE
lines 50-52: Quality of gait, reflected by symmetry in step length and duration of gait phases, tends to change towards greater asymmetry at later stages post stroke [9-11].
AFTER
lines 100-101: Quality of gait, reflected by symmetry in step length and duration of gait phases, tends to change towards greater asymmetry at later stages post stroke [13].
Point 5: Line 76 – the correct term for the cognitive test is the Mini-Mental State Examination. Also, why was a score of 20 selected as the cut-off? Additionally, one exclusion criteria is “mobility deficits significantly limiting and disrupting the patient’s ability to walk”. The participants in this study all have disrupted gait, therefore is this statement intended to mean those that cannot walk independently? May need to be more clear.
Response 5: Thank you for the valuable comment. We have corrected the term "Mini-Mental State Examination". We apologise for the mistake, a score of less than 24 point in the Mini-Mental State Examination is indicative of dementia. In accordance with the Reviewer’s suggestion we have described the exclusion criteria in more specific terms, to make them more clear.
BEFORE
line 76: cognitive impairment (Mini Mental Scale < 20)
lines 75-77: Exclusion criteria were: unstable haemodynamic state, peripheral vascular disease, cognitive impairment (Mini Mental Scale < 20) and mobility deficits significantly limiting and disrupting the patients’ ability to walk.
AFTER
line 125: cognitive impairment (Mini-Mental State Examination < 24)
lines 124-128: Exclusion criteria were: unstable haemodynamic state, peripheral vascular disease, cognitive impairment (Mini-Mental State Examination < 24) and lower limb contractures, difference in the length of the lower limbs exceeding 2 centimetres, osteoarthritis impairing gait, and other orthopaedic, rheumatic and neurologic comorbidities impairing ambulation.
Point 6: Line 92 – what was the distance the participant had to walk in one pass?
In the method, it was described that markers were placed to capture motions at all joints of the lower extremity, including the ankle/foot. The WGS includes items that can be correlated to what it happening at the ankle/foot (e.g. heel strike). Additionally, one reference mentioned in this article [13] reported on the effect of abnormal ankle/foot function on gait symmetry. Why was the ankle/foot not included in this study?
The conclusion reports that the WGS can be a substitute for main aims of 3D assessment, however the results did not support use of any of the WGS components for the assessment of step length and stride length, vital components of the gait cycle.
There are several terms used in this paper to describe the participants of the study (patients, individuals, participants). Recommend modifying for consistent terminology.
Response 6: Thank you for the helpful comment. We have added the distance the participant had to walk in one pass. We fully agree with the comment/question related to inclusion of ankle/foot in the study. Unfortunately, after we performed the 3D tests, we had problems processing and analysing ankle related data for some of the patients, and due to this the joint was disregarded further in the study.
In accordance with the Reviewer’s suggestion, we have changed the conclusion section and have used consistent terminology.
BEFORE
line 92: The patients were asked to walk at a comfortable self-selected speed
lines 246-248: The study confirms that observation of gait using the WGS, which is an ordinal scale, is consistent with the main aims of 3D assessment, therefore the scale can be recommended as a substitute tool in gait assessment.
line 15: gait parameters in individuals after stroke
line 19: Fifty individuals at a chronic stage of recovery
line 21: The study participants’ gait
line 78: participants’ gait
line 122: 50 study participants
line 206: in individuals who have experienced a stroke
line 209: in individuals after stroke
AFTER
lines 142-143: The patients were asked to walk at a comfortable self-selected speed, at a distance of 10 meters
lines 328-330: The study confirms that observation of gait using the WGS, which is an ordinal scale, is consistent with the selected aims of 3D assessment, therefore, the scale can be used as a complementary tool in gait assessment.
line 15: gait parameters in patients after stroke
line 19: Fifty patients at a chronic stage of recovery
line 21: The patients’ gait was
line 128: patients’ gait
line 182: 50 patients
line 275: patients who have experienced a stroke
line 278: in patients after stroke

Round 2
Reviewer 2 Report
Thank you for addressing each of my concerns.